# Fully Automatic Deep Learning Framework for Pancreatic Ductal Adenocarcinoma Detection on Computed Tomography

**DOI:** 10.3390/cancers14020376

**Published:** 2022-01-13

**Authors:** Natália Alves, Megan Schuurmans, Geke Litjens, Joeran S. Bosma, John Hermans, Henkjan Huisman

**Affiliations:** 1Diagnostic Image Analysis Group, Department of Medical Imaging, Radboud University Medical Center, 6500 HB Nijmegen, The Netherlands; megan.schuurmans@radboudumc.nl (M.S.); joeran.bosma@radboudumc.nl (J.S.B.); henkjan.huisman@radboudumc.nl (H.H.); 2Department of Medical Imaging, Radboud Institute for Health Sciences, 6500 HB Nijmegen, The Netherlands; g.litjens@radboudumc.nl (G.L.); John.Hermans@radboudumc.nl (J.H.)

**Keywords:** pancreatic ductal adenocarcinoma, deep-learning, early detection

## Abstract

**Simple Summary:**

Early image-based diagnosis is crucial to improve outcomes in pancreatic ductal adenocarcinoma (PDAC) patients, but is challenging even for experienced radiologists. Artificial intelligence has the potential to assist in early diagnosis by leveraging high amounts of data to automatically detect small (<2 cm) lesions. In this study, the state-of-the-art, self-configuring framework for medical segmentation *nnUnet* was used to develop a fully automatic pipeline for the detection and localization of PDAC lesions on contrast-enhanced computed tomography scans, with a focus on small lesions. Furthermore, the impact of integrating the surrounding anatomy (which is known to be relevant to clinical diagnosis) into deep learning models was assessed. The developed automatic framework was tested in an external, publicly available test set, and the results showed that state-of-the-art deep learning can detect small PDAC lesions and benefits from anatomy information.

**Abstract:**

Early detection improves prognosis in pancreatic ductal adenocarcinoma (PDAC), but is challenging as lesions are often small and poorly defined on contrast-enhanced computed tomography scans (CE-CT). Deep learning can facilitate PDAC diagnosis; however, current models still fail to identify small (<2 cm) lesions. In this study, state-of-the-art deep learning models were used to develop an automatic framework for PDAC detection, focusing on small lesions. Additionally, the impact of integrating the surrounding anatomy was investigated. CE-CT scans from a cohort of 119 pathology-proven PDAC patients and a cohort of 123 patients without PDAC were used to train a *nnUnet* for automatic lesion detection and segmentation (*nnUnet_T)*. Two additional *nnUnets* were trained to investigate the impact of anatomy integration: (1) segmenting the pancreas and tumor (*nnUnet_TP*), and (2) segmenting the pancreas, tumor, and multiple surrounding anatomical structures (*nnUnet_MS*). An external, publicly available test set was used to compare the performance of the three networks. The *nnUnet_MS* achieved the best performance, with an area under the receiver operating characteristic curve of 0.91 for the whole test set and 0.88 for tumors <2 cm, showing that state-of-the-art deep learning can detect small PDAC and benefits from anatomy information.

## 1. Introduction

Pancreatic ductal adenocarcinoma (PDAC) is the most common form of pancreatic cancer, which has the worst prognosis of all cancer diseases worldwide with a 5-year relative survival rate of only 10.8% [1,2]. The incidence of pancreatic cancer is increasing, and it is estimated to become the second leading cause of cancer-related deaths in Western societies by 2030 [2,3]. Patients diagnosed in early disease stages, where the tumors are small (size < 2 cm) and frequently resectable, present a much higher 3-year survival rate (82%) than patients diagnosed in later disease stages where the tumors are larger (17%) [4]. Unfortunately, tumors are rarely found in early stages, and approximately 80–85% of patients present with either unresectable or metastatic disease at the time of diagnosis [1]. Given these statistics, it is clear that early diagnosis of PDAC is crucial to improve patient outcomes, as reversing the stage distribution would more than double the overall survival, without any additional improvements in therapy [5].

Early PDAC detection is challenging, as most patients do not present specific symptoms until advanced disease stages, and screening the general population is cost-prohibitive with current technology [5,6]. Furthermore, PDAC tumors are difficult to visualize in computed tomography (CT) scans, which are the most used modality for initial diagnosis, as lesions present irregular contours and poorly-defined margins [5]. This becomes an even more significant challenge in the initial disease stages as lesions are not only small (<2 cm), but are also often iso-attenuating, making them easily overlooked even by experienced radiologists [7]. A recent study that reconstructed the progression of CT changes in prediagnostic PDAC showed that suspicious changes could be retrospectively observed 18 to 12 months before clinical PDAC diagnosis. However, the radiologists’ sensitivity at identifying those changes, and consequently referring patients for further investigation, was only 44% [8].

Artificial intelligence (AI) can potentially assist radiologists in early PDAC detection by leveraging high amounts of imaging data. Deep learning models, and more specifically convolutional neural networks (CNNs), are a class of AI algorithms especially suited for image analysis and have shown high accuracy in the image-based diagnosis of various types of cancer [9,10,11]. CNNs take the scan as the input and automatically extract relevant features for the diagnostic task by performing a series of sequential convolution and pooling operations.

Clinically relevant computer-aided diagnostic systems should have the ability to both detect the presence of cancer and, in the positive cases, localize the lesion in the input image, with minimal to no required user interaction.

Recently, deep learning models have started to be investigated for automatic PDAC diagnosis [12,13,14,15,16,17]. However, most studies perform only binary classification of the input image as cancerous or not cancerous, without simultaneous lesion localization. Furthermore, the majority of publications do not focus on small, early-stage lesions, with only one study reporting the model performance for tumors with size < 2 cm [15].

In this study, we hypothesize that state-of-the-art deep learning architectures can be used to detect and localize PDAC lesions accurately, especially regarding the subgroup of tumors with a size < 2 cm. We propose a fully automatic deep-learning framework that takes an abdominal CE-CT scan as the input and produces a tumor likelihood score and a likelihood map as the output. Furthermore, we assess the impact of surrounding anatomy integration, which is known to be relevant for clinical diagnosis [7], on the performance of the deep-learning models. The framework performance is validated using an external, publicly available test set, and the results on the subgroup of tumors < 2 cm in size are also reported.

## 2. Materials and Methods

### 2.1. Dataset

This study was approved by the institutional review board (Radboud University Medical Centre, Nijmegen, The Netherlands, CMO2016-3045, protocol version 3, 21 September 2018), and informed consent from individual patients was waived due to its retrospective design. CE-CT scans in the portal venous phase from 119 patients with pathology-proven PDAC in the pancreatic head (PDAC cohort) and 123 patients with normal pancreas (non-PDAC cohort), acquired between 1 January 2013 and 1 June 2020, were selected for model development.

Two publicly available abdominal CE-CT datasets containing scans in the portal venous phase were combined and used for model testing: (1) The training set of the “The Medical Segmentation Decathlon” pancreatic dataset (MSD) from the Memorial Sloan Kettering Cancer Center (Manhattan, NY, USA), consisting of 281 patients with pancreatic malignancies (including lesions in the head, neck, body, and tail of the pancreas) and voxel-level annotations for the pancreas and lesion [18], and (2) “The Cancer Imaging Archive” dataset from the US National Institutes of Health Clinical Center, containing 80 patients with normal pancreas and respective voxel-level annotations [19,20].

The size of the tumors was measured from the tumor segmentation as the maximum diameter in the axial plane.

### 2.2. Image Acquisition and Labeling

The CE-CT scans were acquired with five scanners (Aquilion One, Toshiba (Tochigi, Japan); Sensation 64 and SOMATOM Definition AS+, Siemens Healthcare (Forchheim, Germany); Brilliance 64, Philips Healthcare (Best, The Netherlands); BrightSpeed, GE Medical system, (Milwaukee, WI, USA)). The slice thickness was 1.0–5.0 mm, and the image size was either 512 × 512 pixels (232 images) or 1024 × 1024 pixels (10 images). Images with 1024 × 1024 pixels were resampled to 512 × 512 prior to inclusion in model development.

All images from the PDAC-cohort were manually segmented using ITK-SNAP version 3.8.0 [21] by trained medical students, and were verified and corrected by an abdominal radiologist with 17 years of experience in pancreatic radiology. The annotations included the segmentation of the tumor, pancreas parenchyma, and six surrounding relevant anatomical structures, namely the surrounding veins (portal vein, superior mesenteric vein, and splenic vein), arteries (aorta, superior mesenteric artery, celiac trunk, hepatic artery, and splenic artery), pancreatic duct, common bile duct, pancreatic cysts (if present), and portomeseneric vein thrombosis (if present).

### 2.3. Automatic PDAC Detection Framework

This study uses a segmentation-oriented approach for automatic PDAC detection and localization, where each voxel in the image is assigned either a tumor or non-tumor label. The models in the proposed pipeline were developed using the state-of-the-art, self-configuring framework for medical segmentation, *nnUnet* [22]. All models employed a 3D U-Net [23] as the base architecture and were trained for 250,000 training steps with five-fold cross-validation.

Regions of interest (ROIs) around the pancreas were manually extracted for both the PDAC and non-PDAC cohorts. An anatomy segmentation network was trained to segment the pancreas and the other anatomical structures (refer to the previous section), using the extracted ROIs from the scans in the PDAC cohort. This network was used to automatically annotate the ROIs from the non-PDAC cohort, which were then combined with the manually annotated PDAC cohort to train three different *nnUnet* models for PDAC detection and localization, namely: (1) segmenting only the tumor (*nnUnet_T*); (2) segmenting the tumor and pancreas (*nnUnet_TP*); and (3) segmenting the tumor, pancreas, and the multiple surrounding anatomical structures (*nnUnet_MS*). These networks were trained with two different initializations and identical five-fold cross-validation splits, creating ten models for each configuration. The cross-entropy loss function was used for the PDAC detection networks as it has been shown to be more suitable for segmentation-oriented detection tasks than the soft DICE + cross-entropy loss function, which is selected by default in the *nnUnet* framework [24,25]. Additionally, the full CE-CT scans from the PDAC cohort were downsampled to a resolution of 256 × 256 and were used to train a low-resolution pancreas segmentation network, which was then employed to automatically extract the pancreas ROI from unseen images during inference.

At the inference time, images were downsampled, and the low-resolution pancreas segmentation network was used to obtain a coarse segmentation of the pancreas. This coarse mask was upsampled back to the original image resolution and dilated with a spherical kernel to close any existing gaps. Finally, a fixed margin was applied to automatically extract the ROI, which was the input to the previously described PDAC detection models.

This extraction margin was defined based on the cross-validation results obtained with the PDAC cohort, so that no relevant information was lost while cropping the ROI.

Each of the PDAC detection models (*nnUnet_T, nnUnet_TP*, and *nnUnet_MS*) outputs a voxel-level tumor likelihood map, which indicates the regions of the image where the network predicts a PDAC lesion and the respective prediction confidence. In the case of the *nnUnet_TP* and *nnUnet_MS* networks, a segmentation of the pancreas is also produced. This segmentation was used in post-processing to reduce false positives outside the pancreas by masking the tumor confidence maps so that only the PDAC predictions in the pancreas region were maintained.

After post-processing, candidate PDAC lesions were extracted iteratively from the tumor likelihood map by selecting the voxel with a maximum predicted likelihood and including all connected voxels (in 3D) with at least 40% of this peak likelihood value. Then, the candidate lesion was removed from the model prediction, and the process was repeated until no candidates remained or a maximum of five lesions were extracted. The final output of the framework was a tumor likelihood defined as the maximum value of the tumor likelihood map.

A schematic representation of the inference pipeline from the original image input to the final tumor likelihood prediction is shown in Figure 1.

### 2.4. Analysis

Patient-level performance was evaluated using the receiver operating characteristic (ROC) curve, while lesion-level performance was evaluated using the free-response receiver operating characteristic (FROC) curve. The ROC analysis assesses the model’s confidence whether a tumor is or is not present by plotting the true positive rate (sensitivity) against the false positive rate (1-specificity) at different thresholds for the model output, defined as the maximum value of the tumor likelihood map. The FROC analysis additionally assesses whether the model identified the lesion in the correct location by plotting the true positive rate against the average number of false positives per image at different thresholds for each individual lesion prediction [26,27]. Each 3D candidate lesion extracted from the tumor detection likelihood map was represented by the maximum confidence value within that lesion candidate for the subsequent FROC analysis. A candidate lesion was considered a true positive if the DICE similarity coefficient with the ground truth (calculated in 3D between the whole extracted candidate lesion volume and the tumor ground truth volume) was at least 0.1. This threshold was set in line with well-established previously published studies of the same nature considering other cancer diseases [10,28] as it addresses the clinical need for object-level localization, while also taking into account the non-overlapping nature of objects in 3D.

To compare the three different PDAC-detection configurations, the ten trained models for each were applied individually to the test set. A permutation test with 100,000 iterations was then used to assess statistically significant differences between the area under the ROC curve (AUC-ROC) and partial area under the FROC curve (pAUC-FROC), which was calculated in the interval of [0.001–5] false positives per patient. A confidence level of 97.5% was used to assess statistical significance (with Bonferroni correction for multiple comparisons). The final performance for each configuration was obtained by ensembling the predictions of the ten models.

## 3. Results

The clinical characteristics of the patients in the PDAC cohort are summarized in Table 1. For the non-PDAC cohort, the mean age was 52.3 ± 21.4 (years), and there were 54 female and 69 male patients.

The performances of the three different PDAC detection network configurations on the internal five-fold cross-validation sets are shown in Table 2. At the patient level, *nnUnet_MS* achieves the best performance, with an AUC-ROC of 0.991. Regarding lesion localization performance, the three configurations achieve a similar pAUC-FROC, with *nnUnet_MS* and *nnUnet_TP* performing slightly better than *nnUnet_T*.

The mean ROC and FROC curves obtained on the external test set with each PDAC detection network configuration are shown in Figure 2, with the respective 95% confidence intervals. These curves were calculated using the 10 different trained models (two initializations with five-fold cross-validation) for each configuration. *nnUnet_MS* and *nnUnet_TP* both achieve an AUC-ROC around 0.89, which is significantly higher than *nnUnet_T* (*p* = 0.007 and *p* = 0.009, respectively). At a lesion level, *nnUnet_MS* achieves a significantly higher pAUC-FROC than both *nnUnet_TP* and *nnUnet_T* (*p* < 10^−4^).

The median size of the tumors in the MSD dataset was 2.5 cm (IQR: 2.0–3.2). There were 73 tumors < 2 cm in size in the MSD dataset. Figure 3 shows the patient and lesion level results for each configuration on this sub-set of smaller tumors. At a patient level, the AUC-ROC decreases by about 0.05 for each configuration, when compared to the results obtained on the whole dataset. *nnUnet_MS* and *nnUnet_TP* continued to outperform the *nnUnet_T*, although the differences were not statistically significant at a confidence level of 97.5% (*p* = 0.034 and *p* = 0.077, respectively). Regarding lesion-level performance, the pAUC-FROC for *nnUnet_MS* is still significantly higher than for *nnUnet_TP* and *nnUnet_T* (*p* < 10^−4^ and *p* = 4.8 × 10^−4^, respectively). The results obtained by ensembling the 10 models for each configuration are shown in Table 3.

Figure 4 shows an example of the network outputs of *nnUnet_TP* and *nnUnet_MS* for an iso-attenuating lesion in the neck-body of the pancreas. This lesion is missed by both *nnUnet_T* and *nnUnet_TP,* but is correctly identified by the *nnUnet_MS* model.

## 4. Discussion

In this study, the state-of-the-art, self-configuring framework for medical segmentation, *nnUnet* [22], was used to develop a fully automatic pipeline for the detection and localization of PDAC tumors on CE-CT scans. Furthermore, the impact of integrating the surrounding anatomy was assessed.

A significant challenge of applying deep learning to PDAC detection is that the pancreas occupies only a small portion of abdominal CE-CT scans, with the lesions being an even smaller target within that region. Training and testing the networks with full CE-CT scans would be very resource-consuming and provide a lot of unnecessary information regarding the surrounding organs, distracting the model’s attention from the pancreatic lesion location. In this way, it is necessary to select a small volume of interest around the pancreas, but having expert professionals manually annotate the pancreas before running each image through the network requires extra time and resources, which would significantly diminish the model’s clinical usefulness. To address this issue, the first step in our PDAC detection framework is to automatically extract a smaller volume of interest from the full input CE-CT scan by obtaining a coarse pancreas segmentation with a low-resolution *nnUnet*. To the best of our knowledge, this is the first study to develop a deep-learning-based fully automatic PDAC detection framework and to externally validate it on a publicly available test set.

Previous studies have employed deep CNNs for automatic PDAC detection on CT scans [12,13,14,15,16,17], but only two studies validated their models on an external test set [15,16], with one using the publicly available pancreas dataset. Liu and Wu et al. [15] developed a 2D, patch-based deep learning model using the VGG architecture to distinguish pancreatic cancer tissue from non-cancerous pancreatic tissue. This approach required prior expert delineation of the pancreas, which was then processed by the network in patches that were classified as cancerous or non-cancerous. At a patient level, the presence of a tumor was then determined based on the proportion of patches that the model classified as cancerous. The authors tested this model on the external test set and achieved an AUC-ROC of 0.750 (95%CI (0.749–0.752)) for the patch-based classifier, and 0.920 (95%CI (0.891–0.948)) for the patient-based classifier [15]. On the sub-group of tumors < 2 cm in size, the model achieved a sensitivity of 0.631 (0.502 to 0.747). More recently, Si et al. [16] developed an end-to-end diagnosis pipeline for pancreatic malignancies, achieving an AUC-ROC of 0.871 in an external test set, but validation on the publicly available dataset was not performed.

Our proposed automatic PDAC detection framework achieved a maximum ROC-AUC of 0.914 for the whole external test set and 0.876 for the subgroup of tumors < 2 cm in size. This performance is comparable to the current state-of-the-art for this test dataset [15], but with the advantage of being obtained automatically from the input image, with no user interaction required. Another advantage of our framework is that the lesion location is also identified and so the classification outcomes are immediately interpretable as they directly arise from the network’s segmentation of the tumor. Moreover, the achieved results set a new baseline performance for fully automatic PDAC detection, noticeably improving on the previous best AUC-ROC of 0.871 reported by Si et al. [16].

To the best of our knowledge, this is the first study to assess the impact of multiple surrounding anatomical structures in the performance of deep learning models for PDAC detection. Pancreatic lesions often present low contrast and poorly defined margins on CE-CT scans, with 5.4–14% of tumors being completely iso-attenuating and impossible to differentiate from normal pancreatic tissue [29]. These iso-attenuating tumors are identified only by the presence of secondary imaging findings (such as the dilation of the pancreatic duct) and are more prevalent in early disease stages [7,29]. In clinical practice, surrounding structures such as the pancreatic duct, the common bile duct, the surrounding veins (protomesenteric and splenic veins), and arteries (celiac trunk, superior mesenteric, common hepatic, and splenic arteries) are essential for PDAC diagnosis and local staging [7,29]. However, so far, deep-learning models have focused only on the tumor and noncancerous pancreas parenchyma, not taking the diagnostic information provided by all of the surrounding anatomy into account.

In this framework, the anatomy information was incorporated in the *nnUnet_MS* model, which was trained to segment not only the tumor and pancreas parenchyma, but also several other relevant anatomical structures. The rationale behind this approach was that by learning to differentiate between the different types of tissue present in the pancreas volume of interest, the network could learn underlying relationships between the structures and consequently better localize the lesions. This network was compared to *nnUnet_T*, which was trained to segment only the tumor, and *nnUnet_TP*, trained to segment the tumor and pancreas parenchyma, in order to assess the impact of adding surrounding anatomy.

The results on the external test set show that, at a patient level, there is a clear benefit in adding the pancreas parenchyma when compared to training with only the tumor segmentation, as both *nnUnet_TP* and *nnUnet_MS* achieved a significantly higher AUC-ROC than *nnUnet_T*. There were however no differences in the performances of the *nnUnet_TP* and *nnUnet_MS* networks. Contrastingly, at a lesion-level, there was a clear separation between the three FROC curves both on the whole test set and on the subgroup of tumors < 2 cm in size (Figure 2 and Figure 3), with *nnUnet_MS* achieving significantly higher pAUC-FROC than the two other configurations. This shows that the addition of surrounding anatomy improves the model’s ability to localize PDAC lesions. Figure 4 illustrates the advantage of anatomy integration in the case of an iso-dense lesion that is obstructing the pancreatic duct, causing its dilation. Both the *nnUnet_T* and *nnUnet_TP* models fail to identify this lesion, as there are no visible differences between the tumor and healthy pancreas parenchyma. However, *nnUnet_MS* can accurately detect its location in the pancreatic neck-body following the termination of the dilated duct. By providing supervised training to segment the duct and other surrounding structures, the neural model can better focus on the remaining regions in the pancreas parenchyma, which may explain its ability to detect faint tumors. Furthermore, the multi-structure segmentation provided by *nnUnet_MS* presents useful information to the radiologist that can assist the interpretation of the network output regarding the tumor.

Despite the promising results, there are two main limitations to this study. First, the models were trained with a relatively low number of patients and only included tumors in the pancreatic head, which could be holding back the performance on external cohorts with heterogeneous imaging data. We are currently working on extending the training dataset to incorporate more patients, including tumors in the body and tail of the pancreas, in order to mitigate this issue. Second, training the anatomy segmentation network requires manual labeling of the different structures, which is resource-intensive. To address this problem, we only manually labeled the images from the PDAC-cohort and used self-learning to automatically segment the non-PDAC cohort, which could be introducing errors in training. Like the previous issue, the solution to this problem is to increase the size of the training dataset so that the model can learn better representations of the anatomy and consequently perform higher quality automatic annotations.

## 5. Conclusions

This study proposes a fully automatic, deep-learning-based framework that can identify whether a patient suffers from PDAC or not, and localize the tumor in CE-CT scans. The proposed models achieve a maximum AUC of 0.914 in the whole external test set and 0.876 for the subgroup of tumors < 2 cm in size, indicating that state-of-the-art deep learning models are able to identify small PDAC lesions and could be useful at assisting radiologists in early PDAC diagnosis. Moreover, we show that adding surrounding anatomy information significantly increases model performance regarding lesion localization. Despite these promising results, additional validation with better curated external, multi-center datasets is still required before these models can be implemented in clinical practice.

## Figures and Tables

**Figure 1 cancers-14-00376-f001:**
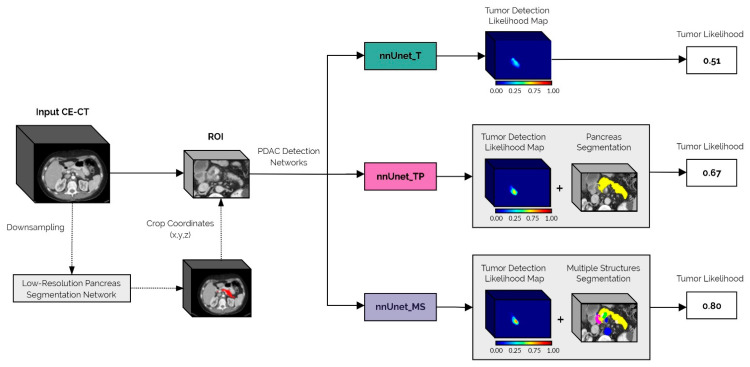
Schematic overview of the proposed automatic PDAC detection framework. The first step in the pipeline is to automatically extract the ROI from the full input CE-CT scan, using the low-resolution pancreas segmentation network. This ROI is then fed to each of the PDAC detection networks: *nnUnet_T, nnUnet_TP,* and *nnUnet_MS*. The final tumor likelihood output is derived from the networks’ tumor detection likelihood maps, which in the case of the *nnUnet_TP* and *nnUnet_MS* models, are post-processed using the automatically generated pancreas segmentation.

**Figure 2 cancers-14-00376-f002:**
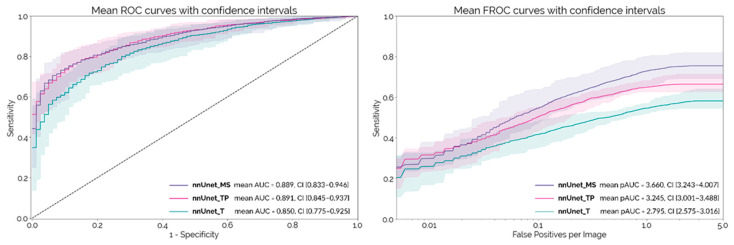
Mean ROC and FROC curves with respective confidence intervals for the external test set.

**Figure 3 cancers-14-00376-f003:**
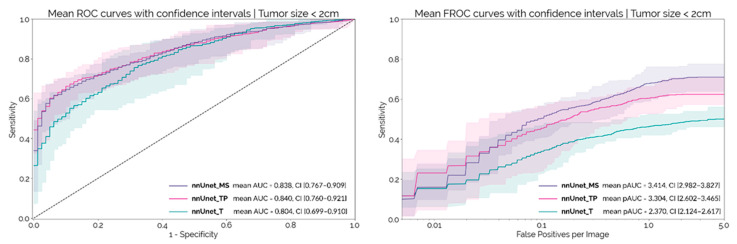
Mean ROC and FROC curves with respective confidence intervals for the external set considering only the subgroup of tumors < 2 cm in size.

**Figure 4 cancers-14-00376-f004:**
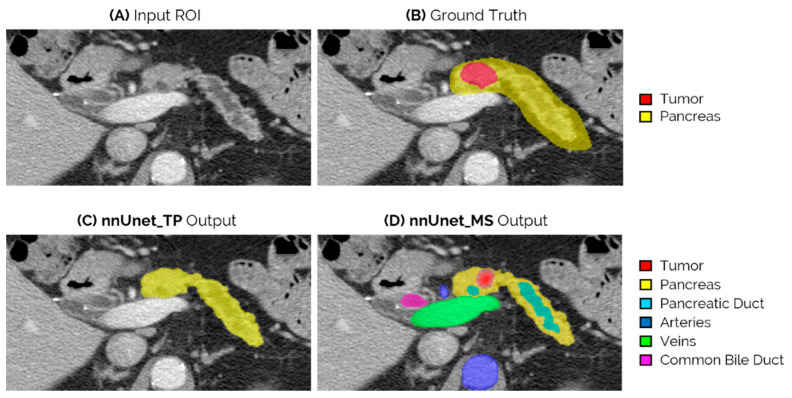
Example of an iso-attenuating tumor from the external test set, which was missed by both the *nnUnet_T* and *nnUnet_TP*, but could be correctly localized by *nnUnet_MS*. (**A**) Slice of the original ROI input; (**B**) ground truth segmentation of tumor and pancreas; (**C**) output of *nnUnet_TP*, which in this case is only the pancreas segmentation as the tumor is not detected; and (**D**) output of the *nnUnet_MS*, which is the segmentation of the detected tumor and surrounding anatomy.

**Table 1 cancers-14-00376-t001:** Clinical characteristics of the patients in the PDAC cohort. Data are mean ± standard deviation or median (interquartile range). The tumor stages are I—locally resectable; II—borderline resectable; III—locally advanced; IV—metastasized.

Clinical Characteristics	
Age (years)	69.2 ± 8.5
Gender (M/F)	67/52
Tumor Stage (I/II/III/IV)	22/21/47/29
Tumor size (cm)	2.8 (2.3–3.7)

**Table 2 cancers-14-00376-t002:** Mean and 95% confidence interval (95% CI) of the area under the ROC curve (AUC-ROC) and partial area under the FROC curve (pAUC-FROC) for the internal five-fold cross-validation for each configuration.

Configuration	Mean AUC-ROC (95%CI)	Mean pAUC-FROC (95%CI)
*nnUnet_T*	0.963 (0.914–1.0)	3.855 (3.156–4.553)
*nnUnet_TP*	0.986 (0.956–1.0)	3.999 (3.252–4.747)
*nnUnet_MS*	0.991 (0.970–1.0)	3.996 (3.027–4.965)

**Table 3 cancers-14-00376-t003:** Ensemble results for the area under the ROC curve (AUC-ROC) and partial area under the FROC curve (pAUC-FROC) from configuration on the whole test set and the subgroup of tumors < 2 cm in size.

Subgroup	Configuration	AUC-ROC	pAUC-FROC
Whole Test Dataset	*nnUnet_T*	0.872	3.031
	*nnUnet_TP*	0.914	3.397
	*nnUnet_MS*	0.909	3.700
Tumors size < 2 cm	*nnUnet_T*	0.831	2.671
	*nnUnet_TP*	0.867	3.289
	*nnUnet_MS*	0.876	3.553

## Data Availability

The data presented in this study are available upon request from the corresponding author dependent on ethics board approval. The data are not publicly available due to data protection legislation.

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
