# Peer review of "Fully Automatic Deep Learning Framework for Pancreatic Ductal Adenocarcinoma Detection on Computed Tomography"

_cancers, 2022, doi:10.3390/cancers14020376_

Round 1
Reviewer 1 Report
The manuscript by Alves et al. reported deep learning based computed tomography detection of pancreatic ductal adenocarcinoma. The authors developed deep-learning based detectection and localization of PDAC assessing the impact of multiple surrounding anatomical structures in the performance of deep learning models for PDAC detection. This manuscript is interesting, but the model has innate limitation in evaluating the performance as to assess pancreatic head tumor. The authors may wish to consider the following comments to improve their manuscript.
Major
1. Results of external validations from two public datasets were presented, but there is no detailedinformation on which part of this public dataset was.
In particular, Memory Sloan Kettering Cancer Center dataset was used as a positive cases, and this paper used 281 pancreatic malignancy cases from total 420 cases from the references (including IPMN, NET, PDAC, etc).
It is necessary to explain by the inclusion criteria how 281 out of a total of 420 data sets were selected, and the details of the test set (size, pathological diagnosis, tumor location, etc..).
2. The authors trained with pancreatic head tumors. Did they also collected also only pancreatic head tumors from public dataset? Did their model can also predict body/tail tumors without training with body/tail tumors?
3. However, it seems that they analyzed only the head tumors, according to the limitation section. Unfortunately, this lead to an another problem. In the case of pancreas, the average length is about 12-15cm, and the head alone is about 3cm. And if you look at the training set, the size of the tumor is 2.3-3.7cm (IQR). Although in the situation of these similar size of tumor and pancreas head itself, the authors set the positiveness criteria as dice similarity coefficient > 0.1. The reviewer think that it would be hard for any lesion that the model detected not to overlap with the tumor, within the limited area of pancreas head.
4. In the case when the model assessed that the tumor is within the pancreas head, any location the model chose will overlap with true tumor. So to evaluate true positiveness, the reviewerrecommends the authors to present the results when the Dice threshold is much higher than 0.1. If the authors first gathered only the head tumors, trained with them, and validated with only the head tumors, the reviewer is afraid that you cannot say if the model matched the location right or not, since the pancreas head is too small organ to claim that.
5. Although the authors describe about detection of early pancreatic cancer tumors (<2cm), the size of tumors from training set is 3-3.7cm (IQR).
6. The authors insist that they evaluated “lesion-level performance”. it is doubtful whether this was not a “lesion-level”but a “slice-level”, which requires a clear explanation. Slice-level is completely different from lesion-level. Let’s say that you found a large tumor that is 10 slices large and missed a small tumor which is 5 slices large (in CT scan), this is a 50% correct rate with a slice-level performance, and the performance will be exaggerated to 66% as lesion-level performance.
Minor
1. The abbreviation CE is used in duplicate, contrast-enhanced and cross-entropy.
2. Page 9 line 287 - English typo. It's not tale, but tail.
Author Response
Dear Reviewer,
Thank you for giving us the opportunity to submit a revised version of the manuscript “Fully Automatic
Deep Learning Framework for Pancreatic Ductal Adenocarcinoma Detection on Computed
Tomography” for publication in the Cancer’s Special Issue on “Pancreatic Cancer: Pathogenesis, Early
Diagnosis, and Management for Improved Survival”. We appreciate the time and effort you dedicated
to providing feedback on our manuscript and are grateful for the insightful comments and valuable
improvements to our paper. We have incorporated your suggestions and addressed your concerns to
the best of our ability. All changes are highlighted in the revised version of the manuscript and below
you will find a point-by-point response to your comments (in red). All page numbers refer to the
revised manuscript file with tracked changes.
Point 1: Results of external validations from two public datasets were presented, but there is no
detailed information on which part of this public dataset was. In particular, Memory Sloan Kettering
Cancer Center dataset was used as a positive cases, and this paper used 281 pancreatic malignancy
cases from total 420 cases from the references (including IPMN, NET, PDAC, etc). It is necessary to
explain by the inclusion criteria how 281 out of a total of 420 data sets were selected, and the details
of the test set (size, pathological diagnosis, tumor location, etc..).
Response 1: We agree the information about the public data set requires further explanation. The
Memory Sloan Kettering Cancer Center dataset used for external validation of the proposed models
was made public in the context of the Medical Segmentation Decathlon Challenge (Antonelli, et al.),
a biomedical semantic segmentation challenge across several different tasks and modalities. In this
challenge, participants were provided with a training set, consisting of 3D images and respective voxel-
level annotations, and a separate test set, consisting of images without the corresponding ground
truth segmentations, which are only held by the challenge organizers. The participants of the
challenge train the models in the labeled training set, and then apply them to the test set, submitting
the obtained segmentation masks to the challenge organizers, who then compare them to the ground
truth and produce the corresponding performance metrics. In the case of the pancreatic cancer
dataset in question, the training set consists of 281 cases with respective annotations for the pancreas
and pancreatic lesions, and the test set consists of 139 unlabelled cases. For this study, we only
considered the cases in the challenge’s training set, as these are the ones that have annotations
available against which we could evaluate the performance of our proposed models. We acknowledge
that this may not have been sufficiently clear in the text and so we made a change to lines 94-100 that
explains the inclusion criteria better. Unfortunately, the information that is provided regarding the
patients in the public dataset is very limited and we do not have access to patient characteristics such
as pathological diagnosis and tumor location. By visual inspection, we can affirm that the dataset
includes tumors arising from all distinct locations in the pancreas (head, neck, body, and tail). We
measure the size of the tumors in this dataset directly from the provided ground truth segmentations
as the maximum diameter in the axial plane, in order to define the subgroup of tumors with size <2cm.
We added the size distribution measured in this way to line 220.
Validating on an independent public data set is the only way to assess the generalizability of our
models and to allow others to compare with our work, an important scientific principle. In the future,
there is the need to collect better curated and complete pancreatic cancer imaging datasets that can
be made public to advance deep learning research in this field. That should indeed include the
different types of pancreatic lesions (IPMN, NET, PDAC, etc). We added a remark to the Conclusions section stating that further external validation with better curated, multi-center pancreatic cancer
datasets is still required before these models could be implemented in clinical practice (lines 349-351).
Point 2: The authors trained with pancreatic head tumors. Did they also collected also only pancreatic
head tumors from public dataset? Did their model can also predict body/tail tumors without training
with body/tail tumors?
Response 2: We agree that our training set consists of patients with tumors in the pancreatic head,
which is the most common location for PDAC (71% of cases) (Erning et al., 2018). Our clinical
experience as a PDAC national expert center is that PDAC in the body and tail of the pancreas tends
to have a larger size than in the pancreatic head (Erning et al., 2018), but we see no major differences
in the appearance of lesions based on their location. In this way, the networks trained on our data set
can learn general, representative features for PDAC to be able to detect lesions in the whole pancreas.
Furthermore, we did not exclude patients from the public external test set based on the tumor
location. This dataset includes a distribution of patients with tumors in the head, neck, body, and tail
of the pancreas. We agree that this was not clear in our original manuscript and have expanded the
information in the Materials and Methods section at line 97 of the revised manuscript. As all locations
were taken into account when testing the models and given the high AUCs obtained in the test set
(>0.8), we can conclude that our models are able to detect tumors in the whole pancreas. In fact,
Figure 4 of the manuscript shows the example of a tumor in the pancreatic neck-body that was
accurately detected and localized by the nnUnet_MS model.
We acknowledge that training the models with tumors arising from a single location in the pancreas
is not ideal and that using a more diversified cohort would very likely improve the models’
performance. We already noted this as a drawback of our study on lines 329-334 of the Discussion
section.
Point 3: However, it seems that they analyzed only the head tumors, according to the limitation
section. Unfortunately, this lead to an another problem. In the case of pancreas, the average length is
about 12-15cm, and the head alone is about 3cm. And if you look at the training set, the size of the
tumor is 2.3-3.7cm (IQR). Although in the situation of these similar size of tumor and pancreas head
itself, the authors set the positiveness criteria as dice similarity coefficient > 0.1. The reviewer think
that it would be hard for any lesion that the model detected not to overlap with the tumor, within the
limited area of pancreas head.
Response 3: As stated in the previous response, we analyzed our models on a public test set
containing tumors localized not only in the head, but also in the neck, body, and tail of the pancreas.
The reviewer correctly points out that the pancreatic head is a small structure and most tumors occupy
a considerable portion of its volume. The suggestion is made that our model is detecting the whole
pancreatic head as a tumor, which would always produce a prediction that overlaps with the ground
truth lesion. Although this may be a valid concern, we do not believe it is the case for our models
because if the networks were simply detecting the pancreatic head instead of the pancreatic tumor
they would misclassify most, if not all, of the negative cases (patients with healthy pancreas) in the
validation and test sets as false positives. This is not the case as can be seen from the high achieved
AUCs (>0.8), which indicates that the models are both sensitive and specific in their classification. By
training, validating, and testing the network using separate cohorts of patients with tumors and
normal pancreas we prevent the network from bluntly associating the whole pancreatic head with the
tumor location, in the subgroup of tumors of the test set that are localized in the pancreatic head.
Regarding the DICE similarity coefficient threshold, the threshold of 0.1 was determined in line with
well-established AI publications of similar nature regarding other cancer diseases such as breast
(McKinney et al., 2020) and prostate (Saha et al., 2021). As ours is the first paper to address
simultaneous pancreatic cancer detection and localization by reporting the free-response receiver
operating characteristic curve (FROC) results, we could not base this threshold definition on previous
literature for the pancreas. Nevertheless, we believe the chosen value of 0.1 is appropriate for our
application. Firstly, the purpose of this study was to develop models that could provide a rough
indication of the tumor location together with the patient-level confidence score, to facilitate
interpretation and assist clinical decision making. We are thus dealing with an object-level localization
task, rather than a semantic segmentation task, which would require precise contour definition and a
higher DICE threshold. Secondly, evaluation and matching are performed entirely in 3D for all models
and, as will be discussed in detail in the response to the next comment, deviations between the model
prediction and the ground truth in 3D have a high impact on the DICE score. By setting the DICE
threshold to a relatively low value of 0.1 we address the clinical need for coarse localization while also
taking into account the non-overlapping nature of objects in 3D. We agree with the reviewer that the
justification for using this threshold could be enhanced in the manuscript and added the previously
discussed points to the Materials and Methods section in lines 176-182.
It is also important to note that although the DICE threshold of 0.1 is applied to classify an extracted
candidate lesion as a true or false positive, all lesions are represented and ranked by a likelihood score
obtained from the tumor likelihood map. It is this likelihood score that indicates the ultimate model
confidence regarding the presence or absence of tumor in that location. We can see the impact of a
wide range of thresholds in this lesion likelihood score by analyzing the computed ROC and FROC
curves (Figures 2 and 3 of the manuscript).
Point 4: In the case when the model assessed that the tumor is within the pancreas head, any location
the model chose will overlap with true tumor. So to evaluate true positiveness, the reviewer
recommends the authors to present the results when the Dice threshold is much higher than 0.1. If
the authors first gathered only the head tumors, trained with them, and validated with only the head
tumors, the reviewer is afraid that you cannot say if the model matched the location right or not, since
the pancreas head is too small organ to claim that.
Response 4: Following the response to the previous comment, we reiterate that the models were not
validated only on tumors in the pancreatic head, but also on tumors in the neck, body, and tail of the
pancreas, which have larger dimensions when compared to the head. Since we tested the models on
tumors from all locations, increasing the DICE threshold would cause the model to classify many
correctly identified lesions as false positives, which is not desirable, especially considering that the
goal of our model is coarse object-level localization and not precise delineation, as previously
discussed.
In our proposed detection framework, the models are trained to produce a voxel-level likelihood map
for the presence of PDAC on the input 3D computed tomography scan, as is shown in Figure 1 of the
manuscript. From this likelihood map, 3D candidate lesions are extracted iteratively as described in
lines 156-160 of the Materials and Methods section. Each of these 3D candidate lesions is then
represented by a likelihood score obtained from the lesion statistics (in our case the maximum
likelihood value within the 3D lesion), which is the score taken into account to assess model
performance regarding detection and localization on the subsequent FROC analysis. The DICE
threshold of 0.1 is used following the standard in deep learning research to assess whether a given
candidate lesion should be classified as a true or false positive, but it has no impact on the final
likelihood outputted by the network for that lesion.
Furthermore, as the reviewer points out, the pancreatic head is a small structure, which makes the
models more likely to over-segment rather than under-segment lesions. This over-segmentation
however is still reflected in the DICE similarity coefficient especially in the 3D setting, which was used
in this paper to calculate the similarity between the model segmentation and the ground truth. In this
3D scenario, even small deviations between the segmentation mask and the ground truth have a
significant impact on the DICE score. This is depicted in Figure 1 of the present document, where we
can see that if a model over-segmented a cubic object by only 3 pixels in each 3D direction the DICE
score (in 3D) between the prediction and the ground truth would be only 0.07. Due to this non-
overlapping nature of objects in 3D, a gross over-segmentation of the lesion in the pancreatic head
would likely cause the DICE coefficient to fall well below 0.1, representing a false positive.
In light of these arguments and the previously referred literature, given the objective of this paper is
to provide an object-level location, the overlap of 0.1 is sufficient to consider a candidate lesion as a
true positive.
Point 5: Although the authors describe about detection of early pancreatic cancer tumors (<2cm), the
size of tumors from training set is 3-3.7cm (IQR).
Response 5: The median size of tumors in our training set is 2.8 cm (IQR: 2.3-3.7 cm). There were 24
cases in the training set with a tumor size <= 2 cm. In this paper, we assess the potential of the
proposed models for early detection of pancreatic tumors by conducting a subgroup analysis
considering the set of tumors with size < 2cm in the external test set. As referred to in line 221 of the
manuscript there were 73 tumors with size < 2cm in the external, publicly available test set, and the
ROC and FROC curves for this subset are shown in Figure 3 of the manuscript. We achieve a maximum
area under the ROC curve of 0.84 for this subgroup, which indicates that although the model was not
exposed to a large number of small cases it can still perform satisfactorily in this subgroup on the
external, unseen test data, showing its potential for early tumor detection.
Figure 1. Schematic representation of an example where both the object (blue) and the prediction (red)
have cubic shapes. The projection in 2D is shown for simplicity.
[Available in PDF Attachment]
Point 6: The authors insist that they evaluated “lesion-level performance”. it is doubtful whether this
was not a “lesion-level” but a “slice-level”, which requires a clear explanation. Slice-level is completely
different from lesion-level. Let’s say that you found a large tumor that is 10 slices large and missed a
small tumor which is 5 slices large (in CT scan), this is a 50% correct rate with a slice-level performance,
and the performance will be exaggerated to 66% as lesion-level performance.
Response 6: In this work, the AI object-level (lesion-level) hit/miss criterium is fully 3D in nature. AI
can exploit all 3D imaging directions and is therefore used in most, current high-impact publications
(McKinney et al., 2020; Saha et al., 2021; Ardila et al., 2019; Yan et al., 2021) as it renders AI agnostic
to the objects' size, morphology, or any other characteristics in 2D/3D. In our approach, the nnUnet
models output a 3D tumor likelihood map directly from the input volume. Then, candidate lesions are
extracted and compared with the ground truth using the DICE similarity coefficient in 3D. In this way,
we do not employ a slice-based approach but consider the 3D lesion detected by the networks as a
whole to evaluate performance. Regarding the example proposed by the reviewer, in our approach if
a model found a large tumor that is 10 slices large and missed a small tumor that is 5 slices large this
would yield a 50% correct rate since the model detected 1 out of 2 lesions (lesion-level performance).
As indicated by the reviewer this performance would be exaggerated 66% (10/15 slices) in a slice-level
approach, which was not adopted in our paper. We rephrased lines 174-179 of the Materials and
Methods section to make this idea more clear.
Point 7: The abbreviation CE is used in duplicate, contrast-enhanced and cross-entropy.
Response 7: We made the correction and removed the abbreviation for cross-entropy in lines 135
and 137.
Point 8: Page 9 line 287 - English typo. It's not tale, but tail.
Response 8: We made the correction in line 333

Reviewer 2 Report
This is a well written and performed study to detect pancreatic ductal adenocarcinoma using deep learning framework.
I consider the topic original and relevant in the field, as there are limited reports on deep learning detection and diagnosis of pancreatic cancer, I believe it adds to the subject area.
One minor concern is that different deep learning models appear to yield vast different segmentation for the tumor, but this has not been assessed (using Dice score, for example). Even with high detection ROC AUC, segmentation error is a concern if the technique is to be used for pre-surgical planning.
While the question on detection of the tumor has been well-addressed, accuracy of segmentation has not been addressed, hence the decision for a minor revision.
Author Response
Dear Reviewer,
Thank you for giving us the opportunity to submit a revised version of the manuscript “Fully Automatic
Deep Learning Framework for Pancreatic Ductal Adenocarcinoma Detection on Computed
Tomography” for publication in the Cancer’s Special Issue on “Pancreatic Cancer: Pathogenesis, Early
Diagnosis, and Management for Improved Survival”. We appreciate the time and effort you dedicated
to providing feedback on our manuscript and are grateful for the insightful comments and valuable
improvements to our paper. We have addressed your concerns to the best of our ability, and below
you will find a point-by-point response to your comments (in red).
Point 1: This is a well written and performed study to detect pancreatic ductal adenocarcinoma using
deep learning framework.
I consider the topic original and relevant in the field, as there are limited reports on deep learning
detection and diagnosis of pancreatic cancer, I believe it adds to the subject area.
One minor concern is that different deep learning models appear to yield vast different segmentation
for the tumor, but this has not been assessed (using Dice score, for example). Even with high detection
ROC AUC, segmentation error is a concern if the technique is to be used for pre-surgical planning.
While the question on detection of the tumor has been well-addressed, accuracy of segmentation has
not been addressed, hence the decision for a minor revision.
Response 1: As the reviewer points out, in our study the accuracy of tumor segmentation is not
assessed (using for instance the DICE score) since semantic segmentation was not the goal of the
proposed models. In this work we use the nnUnet framework to train models geared towards object-
level localization, so that we are able to present the user with both a patient-level pancreatic tumor
likelihood and a coarse location of the lesion, assisting in model interpretation and clinical decision-
making. The main application of our models in the clinical workflow would thus be in the diagnostic
work-up of pancreatic tumors, assisting radiologists in lesions detection and localization, especially in
difficult cases such as small and iso-attenuating lesions. We acknowledge that the proposed models
would not be suitable for tasks such as surgical planning since this would require networks trained to
perform fine semantic segmentation of the tumor and surrounding anatomical structures, which was
not the purpose of this work.

Reviewer 3 Report
An unmet clinical need has been well addressed using artificial intelligence technology. The significance of this work, in comparison with similar studies, is in the fact that it can automatically identify not only the presence of the PDAC lesions but also their location.
The statement of the problem and the proposal of the solution have been formulated in a compelling and seamless way that allows readers to feel the importance of the study. Methodology and research design support predicating the hypothesis. The only slight comment in this section is the lack of ground for including voxels with at least 40% of peak likelihood value in the tumor volume. The presentation of results and use of tables and graphs to support the report are suitable. The conclusion section suitably wraps up the work by putting together the findings of the study and proving the hypothesis. This section however can be reinforced by reflecting on ethics and challenges before deploying the pipeline in clinics. This would be of especial importance for this work if it is intended to be utilized in clinical settings. Additionally, as a comment on this work or suggestion for future study, authors are encouraged to expand on the technical considerations (hardware and software) of launching this framework.
All in all, this paper can be intriguing for readers as it presents a well-structured study that's aiming at addressing a clinical problem with significant importance.
Author Response
Dear Reviewer,
Thank you for giving us the opportunity to submit a revised version of the manuscript “Fully Automatic
Deep Learning Framework for Pancreatic Ductal Adenocarcinoma Detection on Computed
Tomography” for publication in the Cancer’s Special Issue on “Pancreatic Cancer: Pathogenesis, Early
Diagnosis, and Management for Improved Survival”. We appreciate the time and effort you dedicated
to providing feedback on our manuscript and are grateful for the insightful comments and valuable
improvements to our paper. We have incorporated your suggestions and addressed your concerns to
the best of our ability. All changes are highlighted in the revised version of the manuscript and below
you will find a point-by-point response to your comments (in red). All page numbers refer to the
revised manuscript file with tracked changes.
Point 1: An unmet clinical need has been well addressed using artificial intelligence technology. The
significance of this work, in comparison with similar studies, is in the fact that it can automatically
identify not only the presence of the PDAC lesions but also their location.
The statement of the problem and the proposal of the solution have been formulated in a compelling
and seamless way that allows readers to feel the importance of the study. Methodology and research
design support predicating the hypothesis. The only slight comment in this section is the lack of ground
for including voxels with at least 40% of peak likelihood value in the tumor volume. The presentation
of results and use of tables and graphs to support the report are suitable. The conclusion section
suitably wraps up the work by putting together the findings of the study and proving the hypothesis.
This section however can be reinforced by reflecting on ethics and challenges before deploying the
pipeline in clinics. This would be of especial importance for this work if it is intended to be utilized in
clinical settings. Additionally, as a comment on this work or suggestion for future study, authors are
encouraged to expand on the technical considerations (hardware and software) of launching this
framework.
All in all, this paper can be intriguing for readers as it presents a well-structured study that's aiming at
addressing a clinical problem with significant importance.
Response 1: The choice for including all connected voxels with at least 40% of peak likelihood for the
selection of candidate lesions was made empirically based on the training dataset, as this approach
provides the most realistic estimation of the detected lesion. We observed that the specific choice for
this threshold had little impact on the overall performance of our models since the nnUnet with cross-
entropy loss tends to produce sharp prediction borders.
We agree with the reviewer that the Conclusion section could be enriched by adding a remark
regarding the challenges to overcome before deployment in clinical practice. We believe that
additional multi-center external validation with better curated pancreatic cancer datasets is indeed
still required and have added this information to lines 349-351 of the revised manuscript.
